



# CFD analysis of dynamic wind turbine airfoil characteristics in transonic flow using URANS

Maria Cristina Vitulano[1,2], Delphine De Tavernier[1], Giuliano De Stefano[2], and Dominic von Terzi[1]

[1]Aerospace Engineering Faculty, Delft University of Technology, 2629HS Delft, The Netherlands
[2]Engineering Department, University of Campania Luigi Vanvitelli, 81031 Aversa, Italy

**Correspondence:** Maria Cristina Vitulano (M.C.Vitulano@tudelft.nl)

**Abstract.** Modern large wind turbine rotors can encounter airflow at inflow Mach numbers around 0.3 and a Reynolds number of the order of ten million at the blade tip. Our previous study (Vitulano et al., 2025) showed that for these operational conditions, the incompressibility assumption is violated and supersonic flow can occur locally. This follow-up study reports on a numerical investigation of the dynamic behavior of the FFA-W3-211 wind turbine tip airfoil in transonic flow using Unsteady

Reynolds-Averaged Navier-Stokes (URANS) simulations. The simulations are performed for a highly unsteady aerodynamic regime by imposing a dynamic sinusoidal pitching motion across the transonic threshold determined in our previous study. This way, the airfoil is forced to enter and leave the supersonic flow regime. The simulations are conducted by varying the reduced frequency and the inflow Mach number, while keeping the Reynolds number constant at nine million. The choice of non-negligible inflow Mach numbers combined with high Reynolds numbers represents a realistic combination for full-scale

wind turbines, but it is still challenging to be achieved experimentally with the test facilities available nowadays. The dynamic pitching motion is found to lead to the formation of a hysteresis loop with an extent depending on both reduced frequency and inflow Mach number. In particular, it is observed that an increase in one of these two parameters induces an expansion of the hysteresis loop with the consequences of (1) an increase in the magnitude and variability of loads experienced by the airfoil, (2) a delay in the beginning and ending of the transonic flow regime, and (3) the onset of shock waves, that take place at inflow

Mach numbers lower than those estimated under static conditions. Moreover, since the formation of a hysteresis loop implies a range of conditions in which transonic flow can occur, this needs to be better understood and considered when defining any safety margin in the definition of the transonic threshold for turbine design and operation purposes. In general, the study suggests the need to take into account dynamic compressibility effects when predicting aerodynamic loads and performance for next-generation wind turbine rotors.

## 1    Introduction

To address the growing demand for clean energy in recent decades, wind turbine rotors have increased in size, with the largest blades exceeding one hundred meters in length. As a result, the tip speed of current and next-generation wind turbines also increases, reaching the order of one hundred meters per second. In such operational conditions, wind turbine blades experience air flows characterized by inflow Mach numbers around 0.3, where the incompressibility assumption is violated, and even local





supersonic flow could appear at outboard blade sections, leading to the occurrence of shocks, flow separation, and buffeting. This poses a significant challenge to the performance, loads, and fatigue-life predictions of wind turbine rotors.

Several academic investigations have mainly been directed towards analyzing the impact of high-speed flows on large rotors, identifying how aerodynamics could be affected by local variations in air density (see Vitulano et al., 2025, for references). However, it is only rather recently that De Tavernier and von Terzi (2022) pointed out the possibility of transonic flow affecting

the next generation of wind turbines. They analyzed the real operational conditions of the IEA-15MW Reference Wind Turbine (RWT) employing the OpenFAST software and compressibility corrections for the airfoil polars. Their study identified that, near the cut-out wind speed, transonic flow can occur on a portion of the blade tip. In these conditions, the blade operates at high negative incidences, and the flow is strongly accelerated on the airfoil suction side. In addition, local supersonic flow could potentially also appear in off-design conditions at lower wind speeds. These results were more recently corroborated

with experiments conducted by Aditya et al. (2024, 2025) for a limited number of operating conditions, and our numerical analysis reported in Vitulano et al. (2025) over the whole range of relevant parameters.

The methodology proposed by De Tavernier and von Terzi (2022) allowed the establishment of the operating condition for transonic flow occurrence, i.e. the tip speed ratio and pitch angle for a given wind speed or, alternatively, the inflow Mach number and angle of attack. Nonetheless, some uncertainties in their results persisted due to the application of compressibility

corrections, as previously stated by Sørensen et al. (2018). Moreover, this method cannot predict the flow behavior once supersonic flow appears. Vitulano et al. (2025) confirmed the finding provided by De Tavernier and von Terzi (2022) using the more accurate Computational Fluid Dynamics (CFD) approach by solving the Unsteady Reynolds-Averaged Navier-Stokes (URANS) equations. This study highlighted the effects of the Reynolds number in promoting transonic flow, even under normal operating conditions, and the appearance of shock waves as well. Moreover, a "transonic threshold" was found for

three different Reynolds numbers, demonstrating that, for the same Mach number, the occurrence of shocks in the supersonic flow regime depends on the Reynolds number.

In the above analysis, key dynamic effects were not accounted for even though a URANS approach was taken. For example, angle-of-attack changes commonly encountered on wind turbine blades are neglected, resulting in a steady-state analysis. As described in Leishman (2002), wind turbines are affected by interacting flow phenomena that can cause the blade sections to

operate in a highly unsteady flow regime. These effects can cause phenomena like dynamic stall in both design and off-design conditions. Leishman (2002) stated that, under yawed wind conditions, fluctuations in air flow velocity can result in reduced frequencies (introduced below) above 0.1, and even higher inboard on the blade. Also, as the blade passes through the tower shadow, transient changes in the angle of attack occur, potentially resulting in effective values of the reduced frequency that exceed 0.2, corresponding to a highly unsteady regime. In the same work, Leishman outlined the different theories used to

model the effects of unsteady aerodynamics on wind turbines. It is crucial to recognise that the majority of these theoretical methods rely on the assumption of linearity, which limits their validity to low inflow Mach numbers and up to moderate values of reduced frequency. However, in some cases, the assumption of linearity is very difficult to justify, and practical CFD simulations are needed.





Advances have been reported in the literature in characterizing dynamic flow conditions over airfoils. For a range of applications outside of wind energy, Corke and Flint (2015) and Gardner et al. (2023) provide reviews on dynamic stall, including compressibility and the hysteresis of lift and drag as the airfoil enters and leaves the stall regime. Giannelis et al. (2017) describes the different categories of buffeting for transonic (aviation) airfoils. For wind turbine applications, dynamic stall in the incompressible flow regime has received considerable attention in the literature, with the recent work of Chellini et al. (2024, 2025) characterizing dynamic stall for the tip airfoil of the IEA-15MW RWT. However, it remains unclear how wind turbine airfoils behave in transonic flow conditions, especially in the so-called highly unsteady regime.

The present work aims to provide insights into the characterization of the unsteady aerodynamics of the FFA-W3-211 wind turbine airfoil in transonic flow. This airfoil is used at the blade tip of the IEA 15MW RWT (Gaertner et al., 2020) and its successor, the 22MW RWT (Zahle et al., 2024). For this purpose, a dynamic sinusoidal pitching motion is imposed across the transonic threshold on the airfoil, considering the actual operational conditions of a large wind turbine blade section. The main motivation that drove the current computational setup is the dilemma experimentalists face when attempting to achieve realistic operating conditions of large modern wind turbines, as reported in Jung et al. (2022). In wind tunnels currently in use for scientific wind energy research, either the correct (non-negligible) Mach numbers can be achieved albeit at an order of magnitude lower Reynolds number of the order of only one million (see Aditya et al., 2025), or the correct order of magnitude of the Reynolds number but at a much lower Mach number, effectively resulting in incompressible flow, can be attained (see Brunner et al., 2021, for example). Even more often in wind energy research, neither the inflow Mach nor Reynolds number matches full scale. The current CFD analysis is performed by taking into account the dependence on two of the main (dimensionless) parameters characterizing the flow field, specifically the reduced frequency, in a highly unsteady regime, and the inflow Mach number, in a compressible regime. As far as the Reynolds number is concerned, it is maintained at a typical (high) value experienced by tip airfoils of real wind turbine blades.

The manuscript is organized as follows: in Section 2, the numerical methodology is described, highlighting the case studies taken into account. In Section 3, the main findings are presented and discussed. In particular, it is demonstrated that the introduction of a dynamic pitching motion results in the formation of a hysteresis loop, strongly dependent on the value of the reduced frequency and the inflow Mach number. Both the occurrence of transonic flow and shock wave formation are investigated. Finally, some concluding remarks are provided in Section 4.

## 2 Computational model

### 2.1 Case study

The current analysis focuses on the numerical investigation of the unsteady aerodynamic characteristics of the FFA-W3-211 wind turbine tip airfoil used for the IEA-15MW RWT and its successor, the IEA-22MW RWT. These two turbines were designed with input from industry to spearhead research supporting the ongoing development of next-generation wind turbine technologies. For the IEA 15MW RWT, De Tavernier and von Terzi (2022) demonstrated the possibility of the emergence of





transonic flow, and our previous study in Vitulano et al. (2025) established the "transonic threshold" as a combination of inflow Mach number and angle of attack, as reported in Figure 1 (b).

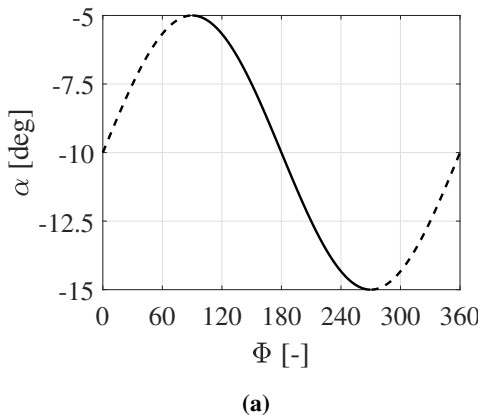
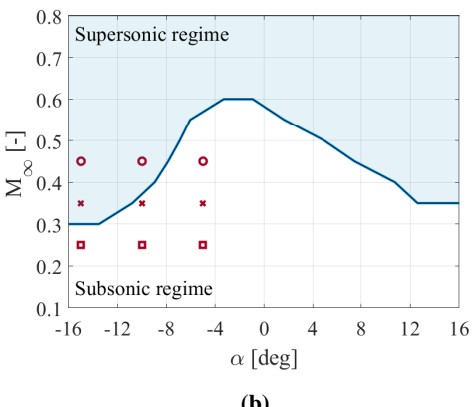

**(a)**  **(b)**

**Figure 1.** Case studies: (a) evolution of the effective angle of attack during one period of oscillation for the pitching airfoil: downstroke (solid line) and upstroke (dashed line); (b) the border of the supersonic flow regimes at $Re = 9 \times 10^6$ (solid blue line from Vitulano et al. (2025)) and the pitching motion mean and boundary values at $M_\infty = 0.25$ (red squares), $M_\infty = 0.35$ (red crosses) and $M_\infty = 0.45$ (red circles).

The sets of computations performed in this work correspond to sinusoidal pitching motion, whose effective angle of attack is given by:

$$a_e(t) = a_m + a_0 \sin\left(\frac{2kt}{t_c}\right) \tag{1}$$

where the mean geometric angle of attack is $a_m = -10°$, and the oscillation amplitude is $a_0 = 5°$. This way, a rotation from an incidence of $-15°$ to $-5°$ is produced across the transonic threshold, forcing the airfoil to enter and leave the supersonic flow region and establishing a transonic regime.

Figure 1 (a) shows the effective angle of attack during a harmonic oscillation period, as a function of the phase angle $\phi = 2kt/t_c$, with $t_c = c/U_\infty$ representing the characteristic time-scale. Here, $c$ is the airfoil chord, and $U_\infty$ is the free-stream velocity. In this picture, the downstroke and upstroke are highlighted, considering the downstroke as that part of the pitching motion in which the airfoil advances into the transonic flow condition, while the upstroke is the motion through which the airfoil moves out of this condition. Note that the unsteady aerodynamics for a pitching motion can involve several physical mechanisms, depending on Reynolds and inflow Mach numbers, and especially the reduced frequency $k = \pi f c/U_\infty$, where $f$ is the physical frequency. All the present computations are performed in the so-called highly unsteady regime, corresponding to $k > 0.2$, according to Corke and Flint (2015).

Specifically, two different sets of simulations are considered for the current CFD analysis that is conducted at a fixed Reynolds number of nine million. First, the effects of introducing a dynamic pitching motion that forces the airfoil to enter and leave the transonic regime are investigated by varying the reduced frequency while fixing the inflow Mach number. This first set of computations is used to identify the value of the reduced frequency that results in the most severe conditions for





the occurrence of transonic flow at very high reduced frequencies, neglecting possibly structural issues that limit experimental analysis. Then, the effect of varying the Mach number is assessed. This would be very challenging to analyze experimentally for a case of particular industrial interest owing to the high Reynolds number. To address this, inflow Mach numbers of 0.25 and 0.35 are selected to represent realistic wind turbine operating conditions. An additional case with an inflow Mach number of 0.45, although slightly beyond typical operational values, is also included to align more closely with experimentally achievable conditions to allow future potential comparisons with experimental data.

For the second set of simulations, the reduced frequency is kept constant while varying the inflow Mach number. The various flow conditions tested are summarized in Table 1.

**Table 1.** Flow conditions for the two different sets of simulations.

|                   | $k$   | $f[Hz]$ | $\omega[rad/s]$ | $M_\infty$ | $Re$            |
| ----------------- | ----- | ------- | --------------- | ---------- | --------------- |
|                   | 0.4   | 16.4    | 185.3           |            |                 |
| $1^{st}$ set      | 0.5   | 22.9    | 144.1           | 0.35       | $9 \times 10^6$ |
|                   | 0.6   | 29.5    | 102.9           |            |                 |
|                   |       | 16.4    | 185.3           | 0.25       |                 |
| $2^{nd}$ set      | 0.6   | 22.9    | 144.1           | 0.35       | $9 \times 10^6$ |
|                   |       | 29.5    | 102.9           | 0.45       |                 |

## 2.2 Numerical setting

The present simulations are carried out by means of the open-source CFD software OpenFoam using the transient solver rhoPimpleFoam, which is particularly suitable for turbulent compressible flow, with the possibility to introduce mesh motion (OpenFoam Foundation, 2016). The current analysis investigates a fully turbulent flow, employing the $k$-$\omega$ Shear Stress Transport (SST) model (Menter, 1994) to account for the effects of unresolved turbulent fluctuations on the resolved two-dimensional mean flow. Free-stream boundary conditions are enforced on pressure, velocity, and temperature fields, while no-slip boundary conditions are imposed at the wall, and adaptive wall functions are used to model the near-wall region. A second-order scheme is used for discretizing the spatial variables, while the first-order implicit Euler scheme is used for the time integration, with a prescribed maximum Courant number of 0.5, to guarantee numerical stability and temporal accuracy. An extensive discussion and validation of the present computational model were provided in our previous work in Vitulano et al. (2025).

The CFD model has been adapted to the current dynamic configuration, enforcing a sinusoidal pitching motion on the airfoil. Practically, the circular computational domain is divided into two concentric subdomains corresponding to an inner and an outer circle, as shown in Figure 2 (a). The outer subdomain is fixed with respect to the reference system, while the inner one, containing the airfoil, rotates around the center of the domain according to the pitching law of Equation 1. The interface between the two subdomains, which is highlighted in red in the figure, is modeled by the cyclic Arbitrary Mesh Interface



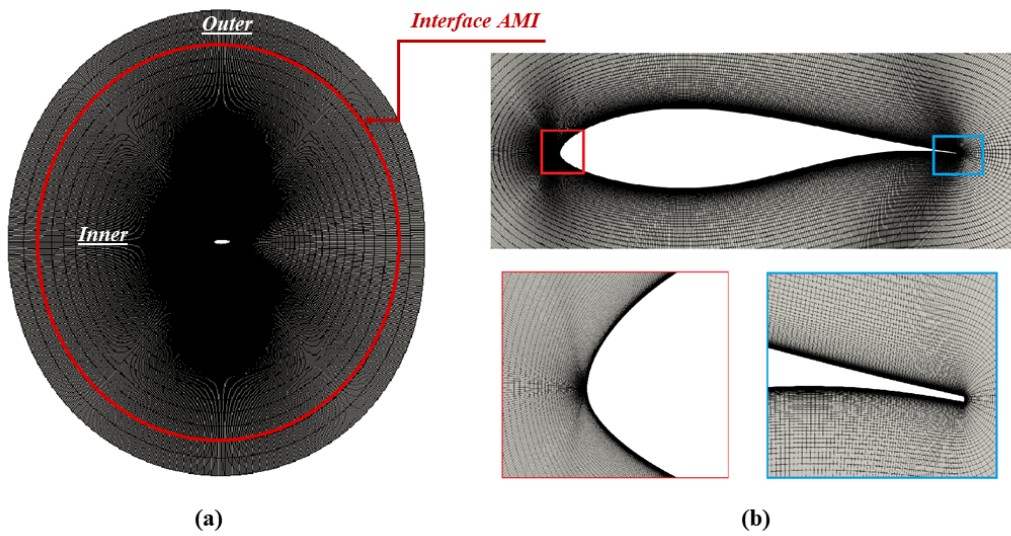

**Figure 2.** Sketch of the computational domain (a), the FFA-W3-211 wind turbine airfoil (b) top, and details of the leading and trailing edges (b) bottom.

(AMI) technique. The cyclic AMI algorithm addresses a periodic interface with a discretization that is equivalent to connecting
135    two adjacent blocks. The two above-mentioned regions can also include non-conforming faces and can even be cyclic. This approach ensures a high convergence rate and robust simulations (Mangani et al., 2017).

## 3    Results

The transonic flow features of a wind turbine tip airfoil strongly rely on the unsteadiness and the associated change in its effective angle of attack (De Tavernier and von Terzi, 2022). This study explores the effect of forcing a wind turbine tip airfoil
140    to enter and leave the supersonic flow regime, and whether this influences the transonic threshold and the occurrence of shock waves. To this end, the impact of a prescribed dynamic pitching motion on the transonic boundary has been assessed in a highly unsteady regime for varying reduced frequency and inflow Mach number.

### 3.1    Reduced frequency dependence

The present section aims to assess the effects of introducing a sinusoidal pitching motion across the transonic threshold of the
145    FFA-W3-211 wind turbine tip airfoil, by varying the reduced frequency in a highly unsteady regime.



### 3.1.1 Transonic flow behaviour for varying frequency

In Figure 3, the minimum pressure coefficient as a function of the effective angle of attack is shown for varying reduced frequency, namely 0.4, 0.5, and 0.6. In aerodynamics, it is established that an increase in velocity leads to a decrease in pressure. As a result, the maximum velocity corresponds to the minimum value of the pressure coefficient. By tracking the minimum pressure coefficient during the pitching motion, it is possible to determine the angles of attack at which transonic flow occurs. Specifically, when the minimum pressure coefficient exceeds a critical value, a transition to a supersonic flow regime is established. A transonic regime is established for all incidences whose minimum pressure coefficient is above the red curve in Figure 3. By inspection of this figure, where the critical level $-C_p = 4.96$ is explicitly indicated, it is possible to discern that the range of angles of attack, for which local supersonic flow occurs, increases with the reduced frequency. Figure 3 also reveals the appearance of a pronounced hysteresis loop, strongly dependent on the reduced frequency, which also appears in the variation of the aerodynamic loads with the angles of attack during the pitching motion, shown in Figure 4. Notably,

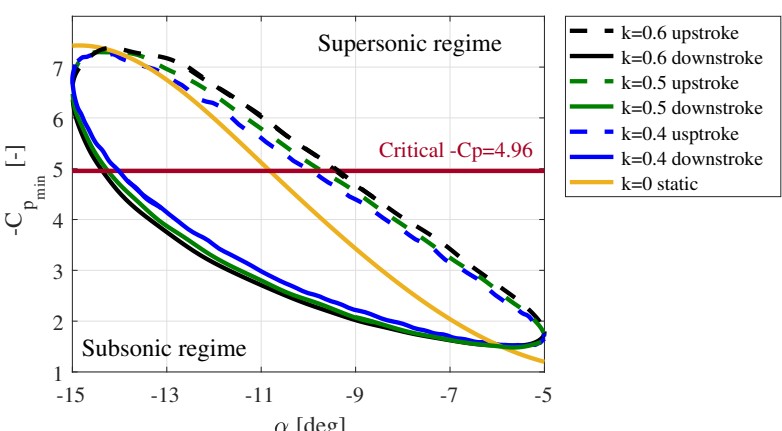

**Figure 3.** Minimum pressure coefficient as a function of the effective angle of attack, for k = 0.4 (blue lines), k = 0.5 (green lines), k = 0.6 (black lines), k = 0 (yellow line): downstroke (solid lines), upstroke (dashed lines); $M_\infty = 0.35$ and Re $= 9 \times 10^6$.

for the downstroke phase defined in the previous section, transonic flow occurs at $\alpha = -14.02°$ for k = 0.4, $\alpha = -14.25°$ for $k = 0.5$, and $\alpha = -14.36°$ for $k = 0.6$, while in the static case, the critical angle of attack is equal to $-10.8°$. A similar delay is also observed when the airfoil is in the upstroke, with the transonic flow disappearing at $\alpha = -9.93°$ for $k = 0.4$, $\alpha = -9.72°$ for $k = 0.5$, and $\alpha = -9.39°$ for $k = 0.6$.

Practically, this numerical analysis demonstrates that increasing the reduced frequency leads to a spread of incidences for which the transonic flow regime is established.

Figure 4 (a) and Figure 4 (b) show the lift and drag coefficients, respectively, as a function of the effective angle of attack during the pitching motion of the airfoil. A moderate hysteresis loop appears for the lift, while a more pronounced hysteresis is established for the drag. Furthermore, the width of the hysteresis loop is strongly correlated with the value of the reduced





frequency, so that higher reduced frequencies correspond to a larger variability of the aerodynamic loads acting on the wind turbine section. In particular, an increase in this parameter results in an expansion of the loop for both lift and drag coefficients, as well as an increase in the risk of establishing a local supersonic flow. In addition, the presence of the hysteresis loop observed in the pressure, lift, and drag coefficients illustrates that defining any threshold with a single value may be misleading

and certainly is prone to high uncertainties, since the existence of a hysteresis loop rather involves the definition of a range of values.

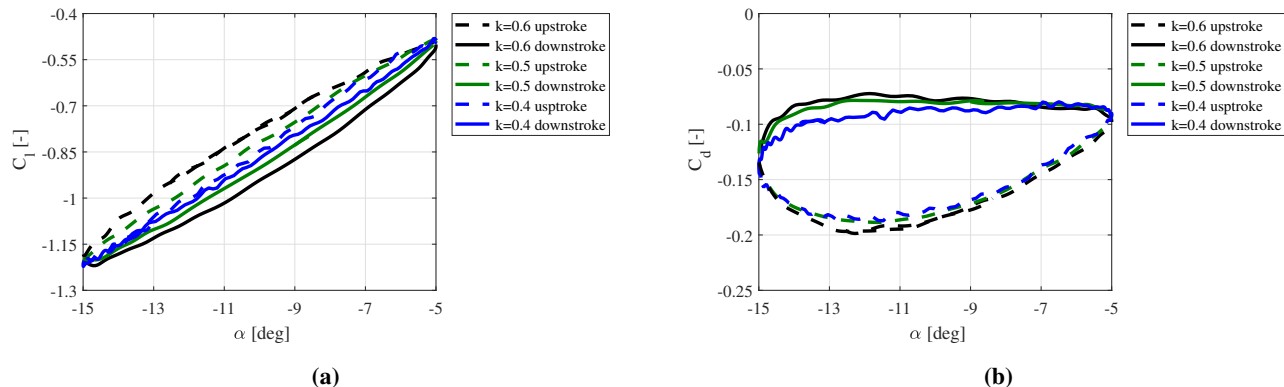

**Figure 4.** (a) Lift coefficient as a function of the effective angle of attack for k=0.4 (blue line), k=0.5 (green line), and k=0.6 (black line); $M_\infty$=0.35 and Re=$9 \times 10^6$: downstroke (solid line), upstroke (dashed line). (b) Drag coefficient as a function of the effective angle of attack for k=0.4 (blue line), k=0.5 (green line), and k=0.6 (black line); $M_\infty$=0.35 and Re=$9 \times 10^6$: downstroke (solid line), upstroke (dashed line).

### 3.1.2    Shock wave occurrence for varying frequency

Following Vitulano et al. (2025), it should be emphasized that establishing a local supersonic flow regime does not necessarily imply the occurrence of shock waves, as also confirmed by experimental findings in (Aditya et al., 2024, 2025). Therefore,

these two physical phenomena must be analyzed separately.

Figure 5 shows the maximum normal Mach number ($M_{n_{max}}$) as a function of the local angle of attack during the pitching motion for various reduced frequencies and an inflow Mach number of 0.35. Note that for this Mach number, our static analysis reported in Vitulano et al. (2025) led to a supersonic flow without shocks. The presence of a shock wave can be detected when the normal Mach number exceeds the value of one. Therefore, Figure 5 illustrates that, for dynamic conditions, shocks could

appear at this lower Mach number. Here, this is the case for all angles of attack for which the maximum normal Mach number is above the red line in the graph.

A significant dependence on the reduced frequency is demonstrated. During the downstroke, shock waves are consistently observed when the pitching angle reaches $\alpha = -15°$. On the other hand, when the airfoil is moving upstroke, an increase in reduced frequency results in a delayed dissipation of shock waves. In particular, shock waves disappear at $\alpha = -12.54°$ for





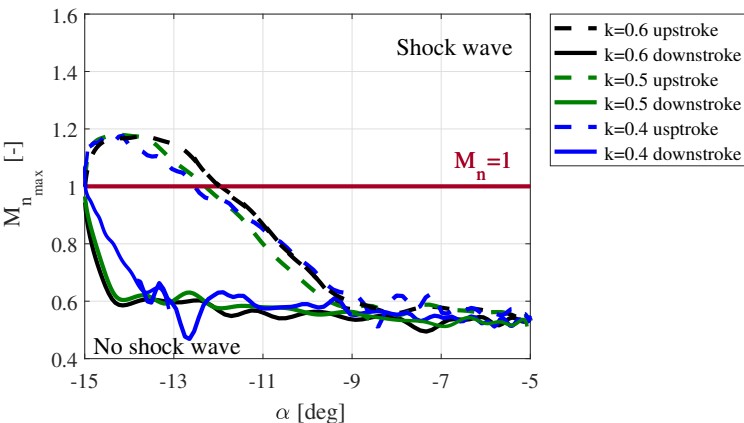

**Figure 5.** Maximum normal Mach number coefficient as a function of the effective angle of attack for k=0.4 (blue line), k=0.5 (green line), and k=0.6 (black line); $M_\infty$=0.35 and Re=$9 \times 10^6$: downstroke (solid line), upstroke (dashed line).

$k = 0.4$, $\alpha = -12.3°$ for $k = 0.5$ and $\alpha = -11.95°$ for $k = 0.6$. As a result, higher reduced frequencies are associated with a wider range of local angles where shock waves occur during the pitching motion.

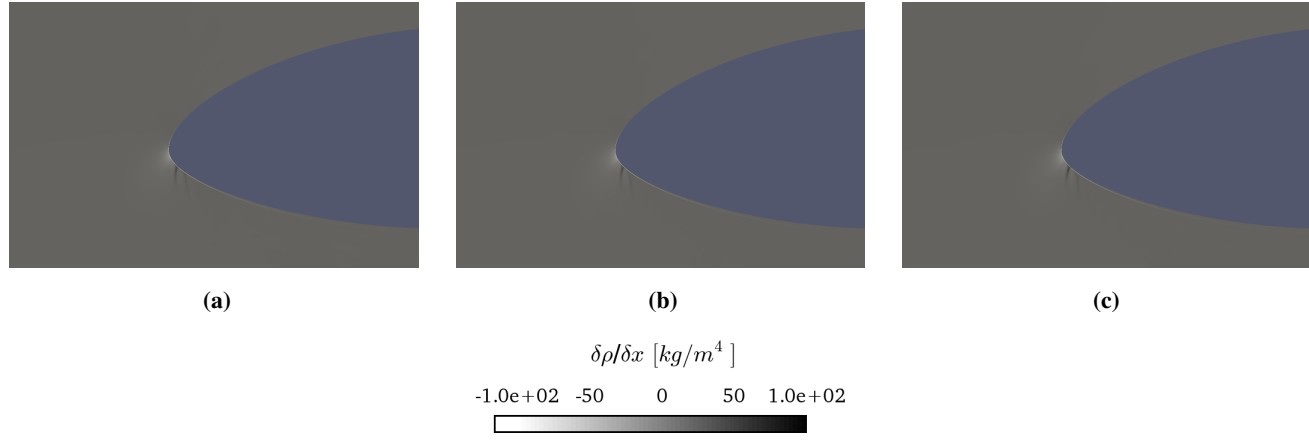

**Figure 6.** Instantaneous (numerical) Schlieren images at (a) k = 0, (b) k = 0.5, and (c) k=0.5 in the upstroke; with $\alpha = -12°$; $M_\infty = 0.35$ and Re = $9 \times 10^6$.

Figure 6 shows the instantaneous (numerical) Schlieren images at different reduced frequencies, for an angle of attack of $-12°$, that is the incidence at which a shock wave should appear for the three frequencies analyzed. A discontinuity in the density gradient is shown near the leading edge for the set of reduced frequencies considered in this work, suggesting that compressibility effects are non-negligible. However, this discontinuity is quite faint, and no significant differences are observed for varying frequency, while, in contrast, the trend of the maximum local normal Mach number reveals a delay in the




dissipation of the shock waves as the reduced frequency increases. This suggests that further analyses, conducted at a higher level of fidelity, will be needed.

## 3.2 Inflow Mach number dependence

The present section aims to assess the effects of varying the inflow Mach number on the transonic flow boundary of the FFA-W3-211 wind turbine tip airfoil, while keeping the Reynolds number at nine million and the reduced frequency at a value of 0.6, i.e. in the highly unsteady regime. In this section, three inflow Mach numbers are investigated, specifically, the values of 0.25 and 0.35 are chosen as representative of realistic operating conditions for this class of turbines. Additionally, an inflow Mach number equal to 0.45, which lies slightly above the typical operational range, is included to facilitate future comparisons

between the present numerical results and forthcoming experimental data.

### 3.2.1 Transonic flow behavior for varying Mach number

Based on the previous discussion, higher frequencies lead to an increased risk for both the onset of transonic flow and the formation of shock waves. Therefore, the following CFD computations are conducted considering the constant reduced frequency of 0.6. Figure 7 shows the predicted minimum pressure coefficient as a function of the effective angle of attack during

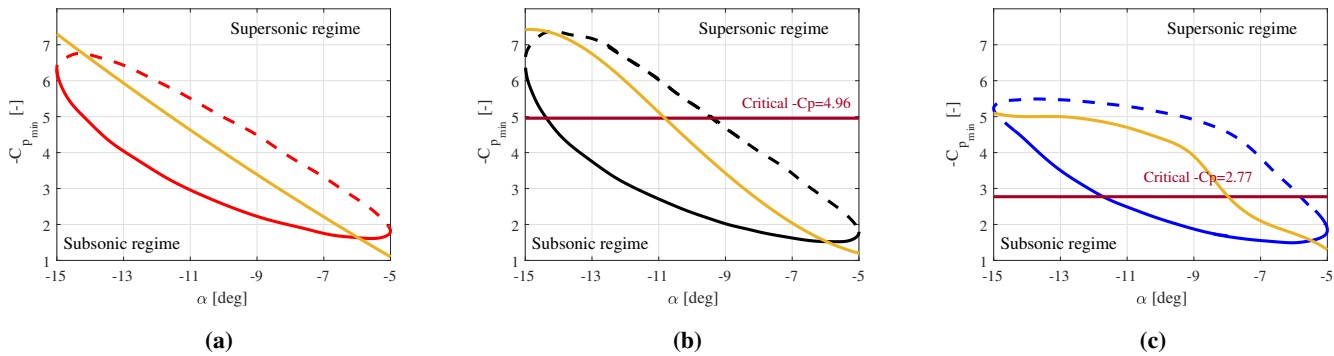

**Figure 7.** Minimum pressure coefficient as a function of the effective angle of attack for $M_\infty = 0.25$ (a), 0.35 (b), and 0.45 (c); k=0.6 and Re=$9 \times 10^6$: downstroke (solid line), upstroke (dashed line).

the pitching motion, for three different inflow Mach numbers that are $M_\infty = 0.25$ (a), 0.35 (b), and 0.45 (c). As mentioned in the previous section, a transonic regime is established for all incidences above the red curve in the picture. The formation of a hysteresis loop is confirmed for all three inflow Mach numbers. Note that the critical $C_p$ is not explicitly shown in Figure 7 (a), being outside of the represented range. For a Mach number of 0.25, the value of the minimum pressure coefficient remains well below the critical level that is $-10.25$, which corresponds to the absence of transonic flow. Considering a Mach number of 0.35,

instead, a local supersonic flow appears when the effective angle of attack surpasses a certain incidence. In particular, a delay in both upstroke and downstroke is observed, compared to the static case where a transonic flow occurs up to an incidence of $-10.8°$ (Vitulano et al., 2025). Practically, this leads to an increase in the range of incidences for which transonic flow occurs,





as shown in Figure 7. Finally, a local supersonic flow also appears for $M_\infty$=0.45. Looking at Figure 7 (c), a plateau for the minimum pressure coefficient appears in the upstroke phase, associated with $\alpha$ approximately ranging from $-15°$ to $-7.5°$; beyond this threshold, a sudden change in the slope of $C_{p_{\min}}$ is observed. This behavior arises as a consequence of the onset stall, which occurs at these incidence and inflow Mach numbers as shown in Figure 8 (a).

**(a)**

**(b)**

**(c)**

**(d)**

**(e)**

**(f)**

*Ma*

0    0.3    0.5    0.8    1

**Figure 8.** Instantaneous streamlines and contour maps of local Mach number; Re = $9 \times 10^6$, k=0.6 and $M_\infty = 0.45$ : (a) $\alpha = -7.5°$ downstroke, (b) $\alpha = -7.5°$ upstroke and (c) $\alpha = -12°$ downstroke, (d) $\alpha = -12°$ upstroke, (e) $\alpha = -15°$ downstroke (f) $\alpha = -15°$ upstroke.





Flow separation arises during the downstroke at an angle of attack of approximately $\alpha = -7.5°$, and the flow remains separated until an incidence of $\alpha = -15°$. Reattachment then occurs when the airfoil is pitching in the upstroke. This behavior is indicative of the onset of a dynamic stall.

It is also crucial to observe an increase in the absolute value of the minimum pressure coefficient, going from $M_\infty = 0.25$ to 0.35, with a subsequent decrease in this quantity at $M_\infty = 0.45$, resulting from the strong compressibility effects experienced by the airfoil in high-speed flow. However, considering the incidences for which the local $C_{p_{\min}}$ exceeds the critical value (that is, points on the curves positioned above the red line in Figure 7 (c)), it is evident that an unsteady flow field with an inflow Mach number of 0.45 is characterized by a broader range of configurations (in terms of local angles of attack) where

a local supersonic flow is established. In fact, Figure 7 (c) shows a strong delay in the occurrence of transonic flow during the downstroke, at $\alpha = -11.72°$, while for the static case, a local supersonic flow appears already at $\alpha = -7.88°$. A delay is also shown for the upstroke phase, for which the airfoil moves out of the transonic regime for an incidence of $\alpha = -5.81°$. Therefore, the present investigation reveals that an increase in the inflow Mach number, for a pitching dynamic motion, leads to an increase in the number of incidences for which transonic flow occurs.

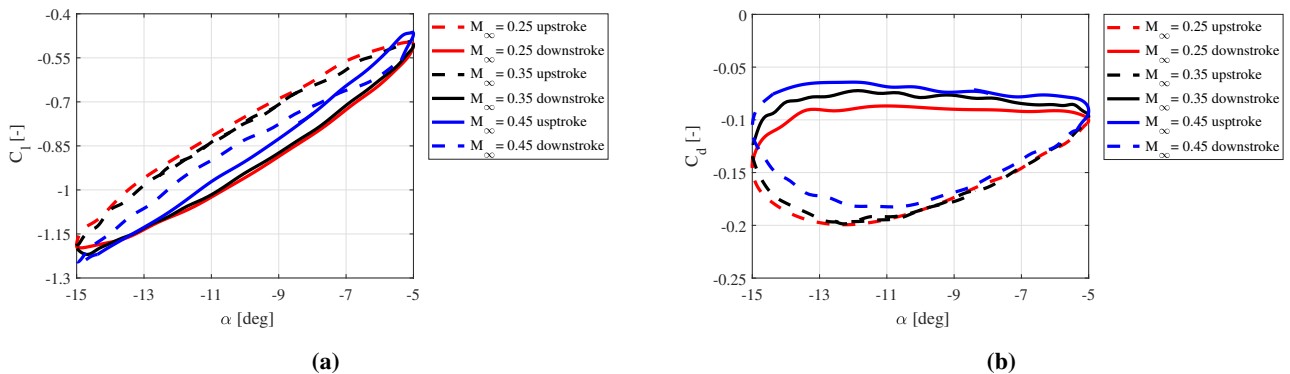

**(a)**                                                                              **(b)**

**Figure 9.** (a) Lift coefficient as a function of the effective angle of attack for $M_\infty$=0.25, $M_\infty$=0.35, $M_\infty$=0.45; k=0.6 and Re=$9 \times 10^6$: downstroke (solid line), upstroke (dashed line). (b) Drag coefficient as a function of the effective angle of attack for $M_\infty$=0.25, $M_\infty$=0.35, $M_\infty$=0.45; k=0.6 and Re=$9 \times 10^6$: downstroke (solid line), upstroke (dashed line).

Figure 9 (a) shows the hysteresis loop of the lift coefficient for varying inflow Mach number. For $M_\infty$ up to 0.35, no substantial changes are observed in terms of aerodynamic load, while a sharp decrease in the amplitude of the hysteresis cycle is observed at $M_\infty = 0.45$, which is consistent with the presence of a plateau in the pressure distribution, as previously discussed. Furthermore, during the downstroke, the lift coefficient increases in magnitude with a constant rate, pitching from an incidence of $\alpha = -5°$ up to $\alpha = -15°$. In contrast, during the upstroke, the coefficient $C_l$ decreases almost linearly until

$\alpha = -7.5°$. Beyond this point, the slope of the curve is increasing considerably, corroborating the hypothesis of the presence of an onset stall at this angle of attack. By inspection of Figure 9 (b), the presence of a hysteresis loop for the aerodynamic drag coefficient is confirmed, with an amplitude appearing unaffected by the inflow Mach number. However, $C_d$ decreases in absolute value with $M_\infty$.



The following discussion focuses only on the analysis of the inflow Mach numbers at which the onset of transonic flow was demonstrated, i.e. $M_\infty = 0.35$ and $0.45$. By inspection of the local Mach number contours plotted in Figure 10 for the angle

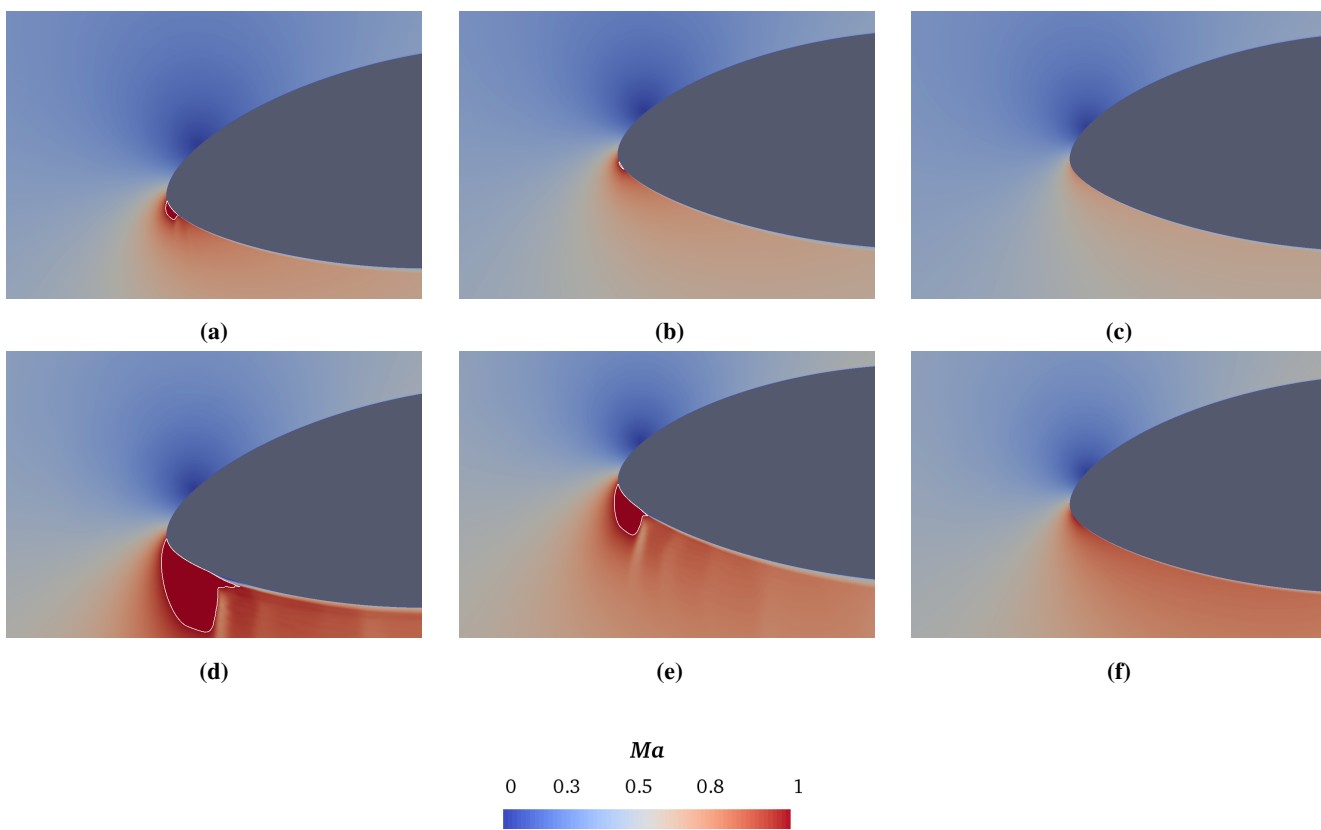

**Figure 10.** Instantaneous contour maps of local Mach number at Re = $9 \times 10^6$ and k=0.6 for varying free-stream velocity: $M_\infty = 0.35$ at (a) $\alpha = -15°$, (b) $\alpha = -10°$ upstroke and (c) $\alpha = -10°$ downstroke; $M_\infty = 0.45$ at (a) $\alpha = -15°$, (b) $\alpha = -10°$ upstroke and (c) $\alpha = -10°$ downstroke. The iso-line corresponds to Ma=0.99.


of attack of $-15°$, a local supersonic flow appears for both Mach numbers. For $M_\infty = 0.35$ (Figure 10 (a)), a small supersonic pocket arises close to the leading edge, while for $M_\infty = 0.45$ (Figure 10 (b)), the supersonic pocket appears enlarged. In the latter case, a clearly defined boundary between supersonic and subsonic flow regions exists, suggesting the presence of a strong shock wave. A similar flow configuration is also observed for $\alpha = -10°$ in the upstroke phase. For $M_\infty = 0.45$, a

significant drop in the velocity field appears, confirming the hypothesis of separated flow (Figure 10 (e)), while a decrease in the inflow Mach number down to 0.35 leads to a significant reduction of the supersonic flow region (Figure 10 (b)). During the pitching downstroke phase, instead, the supersonic region completely disappears for both $M_\infty = 0.35$ and $0.45$, as shown in Figure 10 (c) and Figure 10 (f), respectively. These results clearly suggest that an increase in the inflow Mach number promotes the occurrence of transonic flow for wind turbine tip airfoils.





### 3.2.2 Shock waves occurrence for varying Mach number

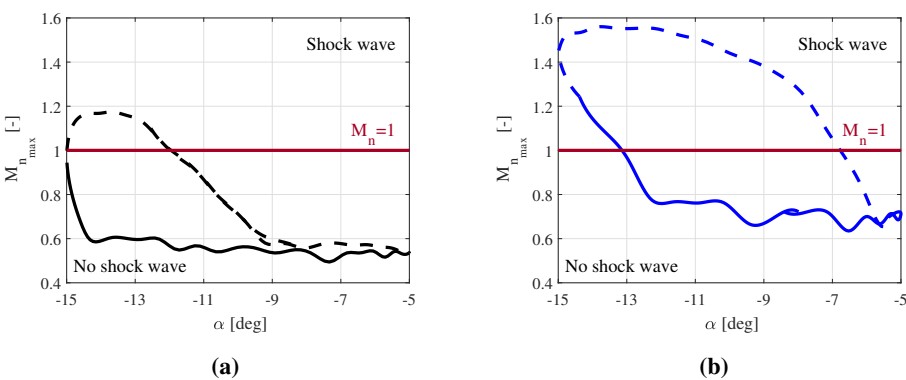

**Figure 11.** Maximum normal Mach number coefficient as a function of the effective angle of attack: (a) $M_\infty$=0.35 and (b) $M_\infty$=0.35; k=0.6 and Re=$9 \times 10^6$, solid line: downstroke, dashed line: upstroke.

The present section focuses on the investigation of the occurrence of shock waves for varying inflow Mach number. Only $M_\infty$= 0.35 and 0.45 are considered, since it was previously demonstrated that transonic flow does not arise at $M_\infty = 0.25$. The trend of the maximum normal Mach number versus the angle of attack, during the pitching motion, is shown in Figure 11. The presence of a shock wave can be detected when the normal Mach number exceeds the value of one, i.e. points above the red line in the graph. The occurrence of shock waves is demonstrated for both $M_\infty = 0.35$ and $0.45$. Notably, even at an inflow Mach number of $0.35$, shock waves occur alongside the establishment of a transonic regime. This represents a novel finding for the present dynamic case, because the previous static analysis performed by Vitulano et al. (2025) found that, at the same Mach number, a transonic flow occurs without the formation of shock waves, even at $\alpha = -15°$. In particular, an increase in the inflow Mach number leads to a rise in the normal Mach number, as well as to an expansion of the range of angles of attack where shock waves appear. Figure 11 (a) shows that at $M_\infty = 0.35$ shock waves are observed only at $\alpha = -15°$ during the downstroke, while Figure 11 (b) reveals that shock waves appear already at $\alpha = -13.15°$. Additionally, a delay in the disappearance of shock waves is noted during the upstroke ascent, at $\alpha = -6.77°$, suggesting that an increase in the inflow Mach number leads to an increase in the incidence at which a shock wave appears. A plateau comparable to that one presented in Figure 7 (c) can also be observed in Figure 11 (b), corresponding to $M_\infty = 0.45$.

Figure 12 presents the instantaneous numerical Schlieren images for an inflow Mach number of either 0.35 or 0.45, at $\alpha = -12°$, for the upstroke phase. At this particular incidence, for an inflow Mach number of 0.35, it can be assumed that the shock wave begins to dissipate. In fact, a slight discontinuity in the density gradient is observed, corroborating this hypothesis. However, when the Mach number is increased to 0.45, a significant increase in this discontinuity is evident, suggesting the presence of a strong shock wave. Furthermore, the separation of the boundary layer flow is observed downstream of the shock. For an inflow Mach number of 0.45, although the flow field is characterized by a large number of configurations in which a shock is present, a significant distinction between the upstroke and downstroke phases is observed, as highlighted in Figure 13,





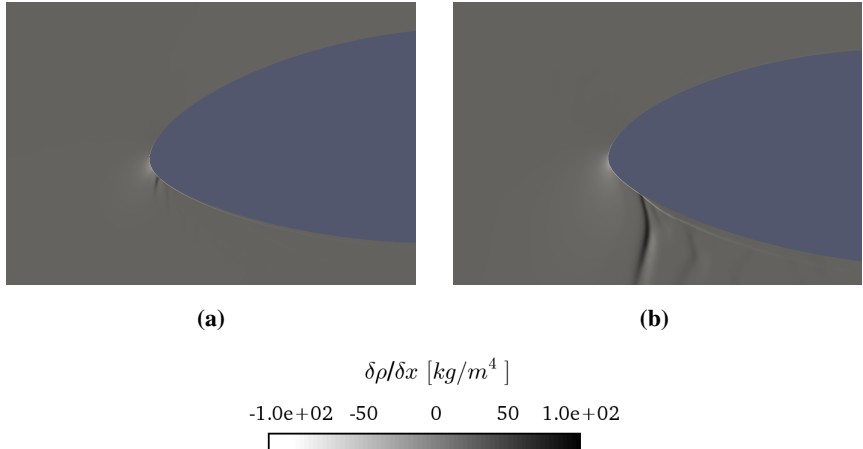

**Figure 12.** Instantaneous (numerical) Schlieren images at (a) $M_\infty = 0.35$ and (b) $M_\infty = 0.45$ in the upstroke; $\alpha = -12°$, $k = 0.6$ and Re = $9 \times 10^6$.

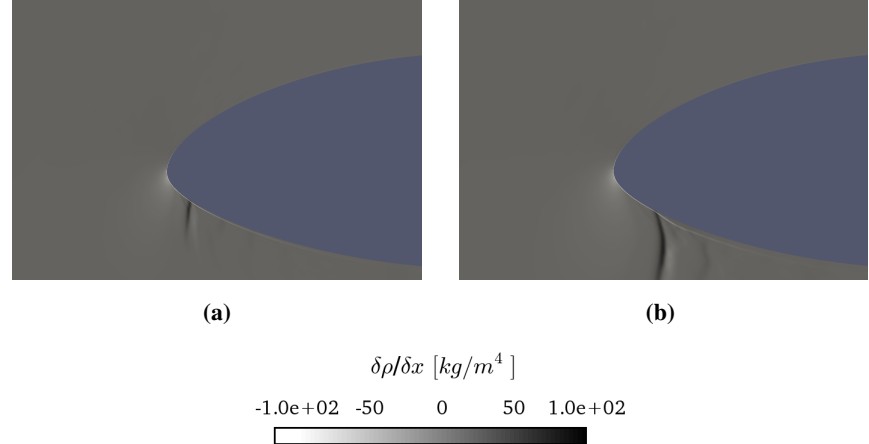

**Figure 13.** Instantaneous (numerical) Schlieren images: (a) $\alpha = -14°$ downstroke and (b) $\alpha = -14°$ upstroke; $M_\infty$=0.45; $k = 0.6$ and Re = $9 \times 10^6$.

which shows the Schlieren images at $\alpha = -14°$. The presence of a shock is observed in both cases, as expected by inspection of Figure 11; however, the discontinuity in the density gradient is significantly greater during the upstroke, suggesting the influence of higher compression loads.





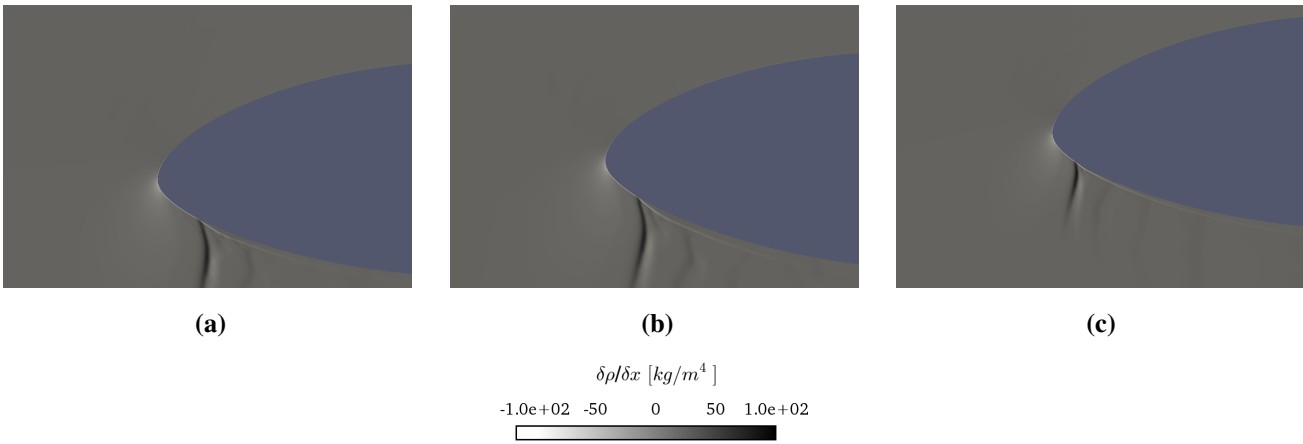

**Figure 14.** Instantaneous (numerical) Schlieren images: (a) $\alpha = -14°$, (b) $\alpha = -12°$ and (c) $\alpha = -8.5°$ upstroke; $M_\infty$=0.45, $k = 0.6$ and Re $= 9 \times 10^6$.

The presence of the plateau depicted in Figure 11 (b) is also supported by the qualitative results shown in Figure 14. At high incidences, specifically at $\alpha = -12°$ and $-14°$, a strong shock wave is observed close to the airfoil leading edge. The shock wave-boundary layer interaction leads to the boundary layer flow separation, which justifies the decrease in the absolute value of the minimum pressure coefficient, compared against the case $M_\infty = 0.35$, where no separation has been observed. Moreover, as the angle of attack reaches $-8.5°$, a weakening of the intensity of the density gradient discontinuity is shown, along with
a reattachment of the boundary layer flow. This fact results in a reduced compression, which is quantitatively reflected in the variation of the slope in the curves shown in Figure 7 (c) and Figure 11 (b), for the minimum pressure coefficient and the maximum normal Mach number, respectively.

## 4    Conclusions

The current numerical investigation provides the unsteady aerodynamic characterization of the FFA-W3-211 wind turbine tip
airfoil in a highly unsteady regime for compressible and transonic flow conditions. The analysis was performed by means of CFD computations using the URANS model previously validated in Vitulano et al. (2025) for steady flow conditions. A dynamic mesh motion was introduced in order to impose a sinusoidal pitching oscillation, forcing the airfoil to enter and leave the supersonic regime. Two different sets of simulations were conducted for varying reduced frequency and inflow Mach number, respectively.
The flow Reynolds number is kept equal to nine million, which is a value typically experienced by a wind turbine tip airfoil at realistic operational conditions for large modern wind turbines. In particular, all the simulations are conducted in the compressible regime, accounting for non-negligible inflow Mach numbers while maintaining a high Reynolds number as well. It is worth stressing that a similar configuration would be very challenging to achieve experimentally.



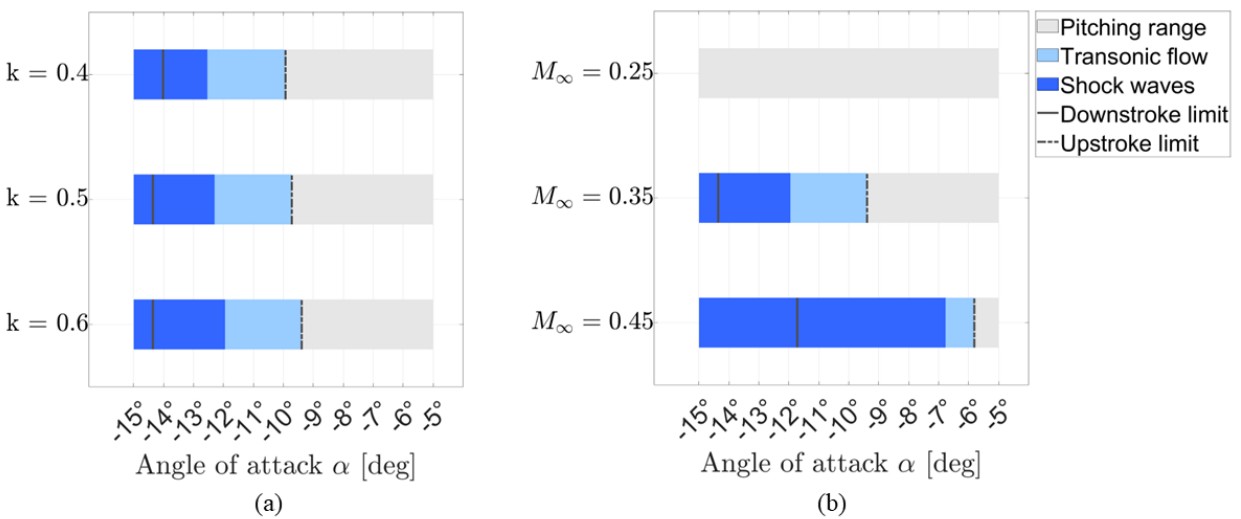

**Figure 15.** Flow regimes during pitching motion as a function of (a) reduced frequency, (b) inflow Mach number, highlighting the angles of attack of the full pitching period (gray), in which transonic flow occurs (light blue), and shock waves appear (dark blue). The transonic limits of the downstroke (solid line) and upstroke (dashed line) are indicated for reference.

This study confirms that flow compressibility plays a crucial role in accurately predicting the aerodynamic performance
and loads experienced by next-generation large wind turbine rotors under real operational conditions in a dynamic regime. In fact, it is shown that the introduction of a harmonic oscillation, with the airfoil entering and leaving the transonic threshold, promotes the onset of a local supersonic flow, as well as the occurrence of shock waves. The flow over the pitching airfoil is also characterized by the establishment of a hysteresis loop, similar but distinct from that associated with dynamic stall. This loop, whose morphology results in being strongly dependent on reduced frequency and inflow Mach number parameters,
affects not only the aerodynamic loads experienced by the airfoil but especially the range of incidences for which transonic flow and shock waves occur. In particular, a delay in transonic flow development is observed, for both downstroke and upstroke phases, caused by an increase of either reduced frequency or inflow Mach number. An incremental change in these quantities leads to an expansion of the hysteresis loop, resulting in more severe loads experienced by the wind turbine tip section.

The present study also reveals that an increase in either reduced frequency or inflow Mach number leads to an increase in the
range of incidences for which local supersonic flow can manifest itself. Shock waves occur already for an inflow Mach number of 0.35, different from previous findings for the static analysis reported (Vitulano et al., 2025), where no such discontinuities were observed for this case. For an inflow Mach number of 0.45, the existence of a strong shock is shown, together with a separation of the boundary layer in a wide range of incidences, during the upstroke phase of the pitching motion. The occurrence of the hysteresis loop also involves the definition of a range of incidences in which transonic flow can occur. Hence,





once the inflow Mach number is fixed, defining only one incidence as a transonic threshold, albeit a useful starting point, would require sufficient safety margins to account for the observed dynamic widening.

All these results are summarized in Figure 15, which illustrates the influence of reduced frequency $k$ (Figure 15 a) and inflow Mach number $M_\infty$ (Figure 15 b) on the flow regime encountered during a pitching motion. The light gray bars represent the full pitching range in terms of angle of attack. The dark blue regions correspond to angles where shock waves are present, while the light blue regions indicate transonic flow without shock formation. Solid black lines denote the downstroke limits, and dashed black lines indicate the upstroke limits. From the analysis of this figure, it is evident that, although an increase in reduced frequency promotes the onset of both transonic flow and shock waves, the inflow Mach number has a significantly stronger influence on the extent and intensity of these flow regimes.

Finally, it is imperative to highlight the limitations of the present analysis. Firstly, the current URANS approach may not be fully adequate for accurately capturing transonic flow phenomena, primarily due to inherent assumptions in turbulence modeling and its interaction with shocks. Actually, the Reynolds-averaging procedure either eliminates or attenuates, in case of strong flow instabilities, a significant amount of flow unsteadiness, while shifting the dominant frequency towards lower values (Fröhlich and von Terzi, 2008). In addition, it is essential to acknowledge that this URANS study only considered fully turbulent flow conditions, whereas laminar to turbulent flow transition could occur on clean wind turbine airfoils. Moreover, the presence of local supersonic flow prompts challenges in the understanding of the influence of shock waves and buffeting on wind turbine performance and longevity, especially in the highly unsteady flow regime. Therefore, it would be essential to evaluate these effects to ensure the efficient operation and durability of future large wind turbines.

Furthermore, given the highly unsteady and three-dimensional nature of the phenomenon under investigation, as well as the necessity to predict dominant frequencies in the flow field to assess aeroelastic instabilities, it is recommended to conduct experiments and/or employ high-fidelity simulations. Considering the limitations associated with the high flow Reynolds number in transonic wind tunnels, future investigations may benefit from utilizing the Large Eddy Simulations (LES) approach, supplied with advanced wall models, or hybrid RANS/LES techniques (Fröhlich and von Terzi, 2008; Salomone et al., 2023).

*Code and data availability.* The code and data can be provided on request by contacting M. Cristina Vitulano.

*Author contributions.* MCV conducted the overall research under the supervision of DDT, GD and DvT. The conceptualisation of the research and the methodology was theorized by MCV, DDT and DvT. The implementation of the computational model and the investigations were carried out by MCV. The results were analyzed and visualised by MCV, DDT, GD and DvT. MCV wrote the original draft, which was reviewed and edited by DDT, GD and DvT.

*Competing interests.* The authors declare that they have no conflict of interest.





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
