# Peer review of "CFD analysis of dynamic wind turbine airfoil characteristics in transonic flow using URANS"

_Wind Energy Science, 2025_

## Author Comment (AC1)

We would like to thank the reviewers for the time and effort they dedicated to reviewing our manuscript. We greatly appreciate their insightful comments and valuable suggestions, which helped to improve the quality and clarity of our work. Below are our point-by-point responses to each of their comments (shown in red). Our replies and the corresponding changes made to the manuscript are highlighted in blue.

**Response to the referee report # 1**

The authors presented "CFD analysis of dynamic wind turbine airfoil characteristics in transonic flow using URANS". This study investigates the effects of aerodynamic performance for unsteady dynamic wind turbines airfoil on compressible and transonic flow condition. A URANS method is used to simulate flowfield of wind turbines airfoil. The findings are presented in manuscript, and the conclusions are supported by the results. The study falls within the scope of wind energy science. It is recommended that the manuscript be accepted after the following comments are addressed.

- **Comment 1**: I think that the authors need to improve the writing of manuscript, mainly the structure manuscript and named of title .

  **Response**: We are not sure what kind of changes in structure and title the reviewer would have liked to see. Without more detailed information, and considering that reviewer 2 states that "the paper is well written," we decided not to make any changes in the structure. However, we adjusted the title by replacing "CFD" with "Numerical", and we could consider dropping "using URANS".

- **Comment 2**: In introduction, the reviewed referenced paper is not enough on compressible study of wind turbine. I suggest that the literature review of this relational study should be added, such as : "Study of air compressibility effects on the aerodynamic performance of the IEA-15 MW offshore wind turbine" in Energy Conversion and Management journal. And "Quantification of air compressibility on large wind turbine blades using Computational Fluid Dynamics" in Renewable Energy journal.

  **Response**: We appreciate that these references were pointed out to us. We agree with the reviewer that the topic of air compressibility in wind turbines is important. However, we believe that a detailed literature review on this aspect is not necessary here, as it was already comprehensively discussed in our previous paper [Vitulano et al., 2025], which is cited in the introduction and on which the present work builds as a natural continuation. Moreover, the articles suggested by the reviewer contain mostly the same references that are already cited in our previous work. Also, these two new publications focus on positive angles of attack and fully subsonic flow, whereas, for the present study, only negative angles of attack and transonic flow are relevant and, hence, investigated (see the explanation given in response to Comment 5). For this reason, we will keep the literature review in the current manuscript concise as it was, while still providing the necessary context for the reader.

- **Comment 3**: For simulation method, the validation case should be considered.

  **Response**: We agree with the reviewer that a validation of the method is very important. However, this was already reported in our previous study [Vitulano et al., 2025], and this is clearly stated in the manuscript (line 128). Specifically, the static model was validated against data available in the literature and, then, extended here to the dynamic case. Unfortunately, no experimental data are currently available for benchmarking the dynamic response, and we acknowledge this as a limitation of the study. Therefore, we are planning to include a more detailed discussion about the model validation and setup in an appendix.

- **Comment 4**: In simulation case, did you make the independent of mesh? I think the authors should do independent of mesh.

  **Response**: We agree with the reviewer that a grid convergence study is important. For the static case, we observed that the solution did not change with further mesh refinement [Vitulano et al., 2025], showing good agreement with the available experimental data. For the present dynamic case, no experimental data are currently available for comparison, so a formal validation study was not possible. Nevertheless, we are willing to include a more detailed discussion on the mesh independence for the dynamic case in the revised manuscript.

- **Comment 5**: In case setting, the authors simulate the range angle within negative angle of attack, if considering within positive angle of attack when the airfoil pitch condition.

  **Response**: We understand that the reviewer is wondering how we chose the range of angles of attack. In the present study, we focused on negative angles of attack because they are the most representative of the actual operating conditions at the blade tip once the rated wind speed is reached, and they also correspond to the incidence range where transonic flow phenomena are likely to occur. The sketch of the power curve on the left of Figure 1 illustrates the typical control strategy of a modern wind turbine. A power curve represents the power output as a function of wind speed: it increases approximately cubically with wind speed up to the rated value, and then it is kept constant through pitch control. In the below-rated regime, the blades are pitched so that the airfoil sections operate at positive angles of attack, close to their aerodynamic optimum. This maximizes the lift-to-drag ratio of the blade sections, thereby enhancing the aerodynamic efficiency of the turbine blades, and enabling the turbine to capture the maximum available power from the incoming wind. In contrast, once the rated wind speed is reached, the pitch control progressively rotates the blades toward a lower angle of attack to reduce the aerodynamic loads and maintain constant power output. As a result, the local angle of attack along the blade sections is progressively reduced, depending on the combination of blade pitch, twist distribution, and local relative wind velocity. At the tip, pitching out the blades will result in negative angles of attack under above-rated conditions, as shown in Figure 1 (right-up). De Tavernier and von Terzi, 2022 considered the operational conditions along 97% of the

[Figure]

Figure 1: Sketch of a power curve (left), sketch of the angle of attack distribution for wind turbine airfoil (right-up), and operational conditions for IEA-15MW RWT tip airfoil from De Tavernier and von Terzi, 2022 (right-down)

blade of the IEA-15MW RWT. Their analysis indicates that local Mach numbers exceeding one can occur at the blade tip close to the cut-out wind speed, highlighting the occurrence of transonic flow. In these operational conditions, the wind turbine tip airfoil is operating at a highly negative angle of attack. For these reasons, we restricted our investigation to negative angles of attack, representing the most relevant regime for the objectives of this study.

- **Comment 6**: For figure 7, it is better that three sub-figures become a figure.

  **Response**: We thank the reviewer for this suggestion. We considered combining the three sub-figures into a single figure; however, this would make the figure overly complex and include too much information, reducing clarity. We therefore believe that keeping them as separate sub-figures best preserves readability and allows for a clearer comparison between the cases.

- **Comment 7**: In figure 12 and 13, can you explain that the small white areas is found, especially figure 12 (b) and figure (13) close pressure face of airfoil? What happen and what reason?

  **Response**: We appreciate the questions of the reviewer. The small white areas observed near the leading edge on the pressure side in Figures 12(b) and 13 correspond to regions with negative density gradients relative to the scale used. In these regions, the flow remains attached, and due to the curvature of the leading edge and compressibility effects at the considered Mach number, the local density decreases slightly along the streamwise direction, generating a negative density gradient. These gradients are captured by the numerical Schlieren visualization, and according to the chosen color mapping, they appear as white regions. To ensure clarity for the reader, we will add this explanation to the revised manuscript.

**Response to the referee report # 2**

The paper addresses a physical phenomenon that concerns modern and future multi-MW wind turbines, that is to say, the possible appearance of supersonic flow over some parts of the blade tip sections, in some specific operating points. URANS simulations are carried out on a pitching FFA-W3-211 airfoil section (representative of the cross section of the tip of an IEA-15MW or IEA 22MW RWT wind turbine blade).

The paper presents the results of six 2D URANS simulations, three with a variable reduced frequency at M = 0.35 and three with a variable inflow Mach number and k = 0.6.

Generally speaking, the paper is well written and of good quality, and the results for the lift and drag coefficients for such applications are relevant. However, additional information would be necessary to give the reader better confidence in the results presented. Details will be provided in the next sections.

\*\*\*\*\*\*\*\*\*\*

Section addressing individual scientific questions/issues ("specific comments")

- **Comment 1**: The airfoil section is shown in Fig. 2 but it would be good to present it at the very beginning of the paper, together with the definition of what is a negative or positive angle of attack (AoA) in this case. I.e., an additional figure would be welcome.

  **Response**: We agree with the reviewer that the discussion regarding the different contributions to the local angle of attack along a wind turbine blade is currently limited. To address this point, we will provide a more detailed explanation of the angle of attack for a wind turbine airfoil in the Case Setting section, and we will include an additional figure, similar to Figure 1, to clearly illustrate the angle of attack convention used in this study.

- **Comment 2**: You should give a little more explanation on the range of angles of attack experienced by the tip of the blade and what does the range that you have selected (-15° to -5°) correspond to. Is it due to the atmospheric boundary layer, to gusts, combination of phenomena? Also, make clear why reduced frequency of 0.4 to 0.6 are relevant to wind turbine applications (by providing simple examples).

  **Response**: We agree with the reviewer that more explanation is needed, as also reviewer 1 struggled with this. The selected range of angles of attack is based on the expected operational conditions of the blade tip of IEA-15MW RWT. In particular, this study is focused on the local angles of attack at which the wind turbine tip airfoil is operating, near the cut-out wind speed, where transonic flow occurs. Figure 1 (right-down) shows that the local angle of attack, when the wind turbine is operating at 25 m/s, varies approximately from $0°$ to $-15°$ at the tip of the IEA-15MW RWT. To focus on the most aerodynamically relevant regimes, we selected a mean angle of attack of $-10°$ with an amplitude of $5°$, resulting in a range from $-5°$ to $-15°$. This choice allows us to consider three representative cases:

$-5°$, corresponding to fully subsonic flow; $-10°$, approximately at the transonic threshold; and $-15°$, where supersonic effects are expected to occur. The reduced frequencies considered in this study were intentionally chosen to explore a highly unsteady aerodynamic regime, which is of critical interest for wind energy applications. While these frequencies are higher than the typical quasi-steady variations experienced in normal operation, they are representative of the rapid transients that can occur at the blade tip due to a combination of factors, including gusts, rotor-induced flow variations, and local accelerations in the boundary layer.

- **Comment 3**: providing the static stall AoA of the airfoil at the studied Re would be nice.

  **Response**: We agree with the reviewer that knowing the static stall angle would be nice. However, currently, there is no static stall information at the Reynolds number considered in this study ($Re = 9 \times 10^6$), from either experiments or high-fidelity simulations (see references in Chellini et al., 2025). We are not sure if a static stall angle at a lower Reynolds number would not be misleading.

- **Comment 4**: section 2.2: be more specific about "second order" scheme (ideally add a table with the main parameters of the fvScheme file, are limiters used?) and give more details about their ability to capture shocks or not by comparison to the literature.

  **Response**: We thank the reviewer for the valuable suggestion. The gradient terms are treated with second-order accurate schemes, with limiters applied to velocity and turbulence quantities to prevent spurious oscillations near steep gradients such as shock waves. Convective terms for momentum are discretized using a linearUpwind scheme, which is robust and accurate in capturing shock-related flow features. Pressure, turbulence, and energy fluxes are treated with a bounded limitedLinear scheme to avoid overshoots and undershoots at discontinuities. Diffusive terms are handled with a corrected Laplacian discretization, improving accuracy on non-orthogonal meshes, and surface-normal gradients are also corrected to enhance the resolution of shock–boundary layer interactions. Finally, a second-order linear interpolation scheme is employed to provide a smooth representation of the flow variables in regions away from discontinuities. This combination of schemes provides a good compromise between accuracy and robustness, ensuring reliable shock resolution without introducing excessive numerical oscillations [Moukalled et al., 2015]. For completeness, a detailed summary of all discretization choices and their rationale will be included in an appendix in the revised manuscript.

- **Comment 5**: Even if you have done it in a previous paper, you should give a minimum set of information about the mesh generation and verification study in this paper (you can still refer to the other one for the details). We need to know the y+ distribution on the airfoil, and to get confidence in the mesh refinement based on a new figure. You should also mention the tool used to generate the mesh.

  **Response**: We thank the reviewer for this comment. We agree that a more extensive discussion of the numerical model is needed to ensure clarity and confidence in the results.

In the revised manuscript, an appendix will be added to provide a more comprehensive description of the numerical setup.

- **Comment 6**: you do mention that experimental data do not exist for such Mach and Reynolds number but some validation with an "as similar as possible" case would help gain confidence in your results. If it is in your earlier.

  **Response**: The reviewer has raised an important point here. We fully acknowledge the significance of validation in building confidence in numerical results. For the present study, however, comparison with existing experimental data would not be particularly meaningful, due to the combination of very high Reynolds numbers and compressible Mach numbers considered, which are beyond those available in the literature. In our previous work, validation was carried out by varying one parameter at a time, allowing the assessment of trends in the flow. Additionally, part of that study employed a comparison with the compressibility correction approach, which, although considering the low-fidelity, provided useful insight into the expected flow behavior. Nevertheless, we agree that a more extensive discussion of the numerical model is warranted, and in the revised manuscript, an appendix will be added to provide a comprehensive description of the numerical setup, enabling readers to fully assess the methodology.

- **Comment 7**: in section 3.1.1 (varying k), you often mention that k has a "strong" effect on the results or that the results have a "significant dependence" on k. This is not what I see in the figures. And actually, it would be interesting to see the mesh sensitivity study to evaluate the relevance of the "small" differences observed. Figures show that there is an effect, but not that strong as far as I see. Maybe you could be more balanced in your writing.

  **Response**: Thank you for pointing out this issue. We agree that the wording regarding the influence of the reduced frequency could be perceived as overly strong. In the revised manuscript, we will adjust the text to provide a more balanced discussion of the results.

- **Comment 8**: At some point you mention that it is possible to observe a supersonic flow without the presence of shock wave but then you mention M = 1 as a criteria for the presence of shock wave. Can you give a little more details on this point?

  **Response**: We thank the reviewer for this comment. This point is precisely what we intended to convey. A flow is considered transonic when a local supersonic region develops, so when the local Mach number exceeds 1 ($M > 1$). However, the presence of a supersonic region does not necessarily indicate the formation of a shock. To identify shocks, we adopt an additional criterion: a shock is present where for the normal Mach number, defined as $M_n = \mathbf{V} \cdot \nabla p / (a|\nabla p|)$, it holds $M_n > 1$. This approach has been already described in detail in our previous work [Vitulano et al., 2025]. In the revised manuscript, this point will be clarified and highlighted more explicitly in the text.

- **Comment 9**: section 3.2: can you explain how you keep Re constant? variable c or variable viscosity?

  **Response**: We thank the reviewer for this comment. In the present study, the Reynolds number is kept constant by adjusting the fluid viscosity.

- **Comment 10**: you mention flow separation in some cases but we have no figure backing this. It would be good to show the wall shear stress (x-component) to give more details about the flow recirculation on the airfoil.

  **Response**: We thank the reviewer for this suggestion. We agree that additional information on flow separation would be useful, and the wall-shear stress plot will be included in the revised manuscript.

- **Comment 11**: line 248-249: seems quite logical.

  **Response**: We thank the reviewer for this comment. While the point may seem straight-forward from an aerodynamic perspective, we feel it is important to emphasize it in the manuscript. Given that the journal focuses on wind energy, not all readers may have a background in aerodynamics, and providing such clarifications helps make the discussion more accessible and clear.

- **Comment 12**: number of periods of oscillation simulated to ensure convergence?

  **Response**: We thank the reviewer for this comment. In the present simulations, twenty oscillation cycles were performed to ensure convergence, so that a fully periodic state was reached. In the revised manuscript, we will explicitly mention this to provide clarity and confidence in the results.

- **Comment 13**: What is the difference between the "local" and the "effective" angle of attack? You use these two words all along the article without explaining the difference (if any). If there is no difference, it would be more clear if you keep only one way to refer to it.

  **Response**: Thank you for your comment. In the present study, there is no difference between the "local" and the "effective" angle of attack; the two terms have been used interchangeably. To improve clarity, in the revised manuscript, we will adopt a consistent terminology throughout the text.

- **Comment 14**: it would be nice if the conclusion can contain a short paragraph that would link the results obtained with the expected outcome on the performance of a wind turbine: what is the effect of properly taking into account the compressibility effect? Cd seems to decrease when M increases (Fig. 9) and Cl does not seem too affected... the effect is not so clear and would benefit from your analysis.

  **Response**: We thank the reviewer for this insightful comment. We agree that including a discussion linking the results to the expected impact on wind turbine performance would be

very valuable. In the revised manuscript, a short paragraph will be added to the conclusions to explicitly address this point.
* * *
Compact listing of purely technical corrections at the very end ("technical corrections": typing errors, etc.)

- **Comment 1**: Fig. 1: x-axis of Fig. (a) =¿ double check the expression of PHI and the variation of it from 0 to 360 (°?).

  **Response**: We thank the reviewer for pointing out this issue. In the manuscript, the phase was originally defined in a dimensionless form:

$$\phi = 2k\frac{t}{t_c} \quad [\text{rad}] \tag{1}$$

  which naturally varies between 0 and $2\pi$ over one oscillation cycle. However, for an easier interpretation of the pitching motion, the phase was reported in degrees, ranging from 0 to 360°. To avoid ambiguity and ensure consistency, in the revised manuscript, we will explicitly define the phase in degrees, namely:

$$\phi = \left(2k\frac{t}{t_c}\right)\frac{180}{\pi} \quad [\text{deg}] \tag{2}$$

  and we will use this convention throughout the text and figures.

- **Comment 2**: Fig. 2: it is difficult to get a good quality figure of a mesh and I have troubles properly seeing the mesh details on my printed version. Tools like fluidFoam (https://fluidfoam.readthedocs.io/en/latest/) allow to export OpenFOAM meshes in vectorized format. You may consider trying something like that. Also, keep an aspect ratio of 1:1 for the computational domain (left part of Fig. 2) and explain why keeping only 2 cells in the outer (stator) region.

  **Response**: We thank the reviewer for the constructive suggestions regarding the mesh visualization. We agree that improving the quality and readability of the mesh figure would be beneficial. In the revised manuscript, we will improve the quality of Fig. 2, following the reviewer's suggestion. Regarding the discretization of the outer region, only two cells were retained because this part of the domain has a negligible influence on the flow around the airfoil, and such resolution is sufficient to accurately impose the far-field boundary conditions. To further minimize any potential numerical artifacts, the interface between the outer and inner regions was positioned as far as possible from the airfoil, ensuring that the flow in the region of interest remains well resolved and unaffected by the coarser mesh in the outer region. This approach also allows for a significant reduction of the overall computational complexity, while maintaining the accuracy in the region of primary interest.

- **Comment 3**: Fig. 6: a closer view would help seeing better the differences between the shocks + (b) is not well defined in the caption (downstroke?).

  **Response**: We thank the reviewer for this comment. We acknowledge that the caption for panel (b) was incorrectly described, and in the revised manuscript, it will be corrected to clearly indicate the flow condition. Moreover, although we initially maintained the same zoom level throughout the paper for consistency and to facilitate comparison, we agree that a closer view of Fig. 6 would better highlight the differences between the shocks. Accordingly, a zoomed-in version will be added in the revised manuscript.

- **Comment 4**: Fig. 7: make clear what the orange line correspond to (fixed airfoil configuration, I guess)

  **Response**: We thank the reviewer for this comment. The orange line in Fig. 7 represents the fixed airfoil configuration. In the revised manuscript, the caption will be adjusted to clearly indicate this fact.

- **Comment 5**: Fig. 10: problem in the labels in the caption for the lower row (d,e,f).

  **Response**: We thank the reviewer for pointing this out. The labels in the caption for the lower row of Fig. 10 were indeed incorrect. In the revised manuscript, the caption will be corrected to properly reflect panels (d), (e), and (f).

- **Comment 6**: Fig. 11: issue in the labels written in the caption (twice the same value M = 0.35, while second one should be M = 0.45, I guess)

  **Response**: We thank the reviewer for pointing this out. The caption of Fig. 11 incorrectly reports the Mach number for panel (b). In the revised manuscript, it will be corrected to indicate $M_\infty = 0.45$, while panel (a) remains $M_\infty = 0.35$.

**References**

1. Vitulano, Maria Cristina, et al. "Numerical analysis of transonic flow over the FFA-W3-211 wind turbine tip airfoil." Wind Energy Science 10 (2025): 103–116.

2. De Tavernier, Delphine, and Dominic Von Terzi. "The emergence of supersonic flow on wind turbines." Journal of Physics: Conference Series. Vol. 2265. No. 4. IOP Publishing, 2022.

3. Chellini, Simone, Delphine De Tavernier, and Dominic von Terzi. "The experimental characterisation of dynamic stall of the FFA-W3-211 wind turbine airfoil." Wind Energy Science Discussions 2025 (2025): 1-26.

4. Moukalled, Fadl, Luca Mangani, and Marwan Darwish. "The finite volume method." The finite volume method in computational fluid dynamics: An advanced introduction with OpenFOAM® and Matlab. Cham: Springer International Publishing, 2015. 103-135.